# Bacterial Pathogens and Antimicrobial Susceptibility Patterns of Urinary Tract Infections in Children during COVID-19 2019–2020: A Large Tertiary Care Center in Saudi Arabia

**DOI:** 10.3390/children10060971

**Published:** 2023-05-30

**Authors:** Ibraheem Altamimi, Abeer Almazyed, Sami Alshammary, Abdulaziz Altamimi, Abdullah Alhumimidi, Raed Alnutaifi, Mohammed Malhis, Abdullah Altamimi

**Affiliations:** 1College of Medicine, King Saud University, Riyadh 11461, Saudi Arabia; zxashzx@gmail.com (A.A.);; 2Microbiology Department, King Fahad Medical City, Riyadh 11525, Saudi Arabia; aalmazyed@kfmc.med.sa; 3Palliative Care, King Fahad Medical City, Riyadh 11525, Saudi Arabia; sami.alshammary@health.sa; 4College of Medicine, King Saud Bin Abdulaziz University for Health and Sciences, Riyadh 11481, Saudi Arabia; altamimi10232@ksau-hs.edu.sa; 5Pediatric Emergency and Medical Toxicology, King Fahad Medical City, Riyadh 11525, Saudi Arabia; tamimi7a@gmail.com

**Keywords:** antimicrobial susceptibility, pediatric urinary tract infection, extended-spectrum beta-lactamase, empirical treatment, antimicrobial resistance, COVID-19

## Abstract

Background: One of the most prevalent bacterial infections in children is urinary tract infection (UTI), which has become a major concern with increasing resistance of the pathogens to the routinely used antimicrobial agents. The aim of the study is to determine the antimicrobial susceptibility patterns of pediatric UTI-causing pathogens, including ESBL-producing bacteria, in Saudi Arabia. Methods: This cross-sectional retrospective study was conducted to ascertain the frequency of isolation and the antimicrobial resistance pattern of uropathogens among children aged 0–15 years. The data from the urine cultures was collected during 2019–2020 at the King Fahad Medical City, a major tertiary hospital in Riyadh, Saudi Arabia. A total of 1022 urine samples from patients diagnosed with urinary tract infections (UTIs) were collected for this study. Microbial species present in the samples were cultured and identified using standard biochemical techniques. To assess the resistance of these strains to antimicrobial drugs, an in vitro method was employed, and the criteria set by the Clinical Laboratory Standard Institute (CLSI) were followed. In addition, a double-disc synergy test was conducted to identify strains of *E. coli* that produce extended-spectrum beta-lactamase (ESBL). Results: The predominant pathogens were *E. coli* (58.6%), followed by *Klebsiella* sp. (23.9%). *E. coli* isolates were more sensitive to meropenem and ertapenem in 99.2% of cases, followed by amikacin (99%). *Klebsiella* sp. were sensitive to amikacin in 97.1% of cases, followed by meropenem and ertapenem (92.2% in both). The highest sensitivities of antimicrobials toward ESBL were for meropenem and ertapenem (100% in both), followed by amikacin (99%). Conclusions: Our study recommends using local antibiotic sensitivity data for empirical UTI treatment. Amikacin, ertapenem, and meropenem are effective intravenous options. Cephalosporin, cefuroxime, amoxicillin/clavulanic acid, and nitrofurantoin are suitable oral choices. No significant changes in antimicrobial susceptibility were observed during the COVID-19 pandemic. Further research is needed to assess potential pandemic-related alterations.

## 1. Introduction

Pediatric urinary tract infection (UTI) is a prevalent disease in healthcare institutions and communities, in both outpatients and inpatients [1]. More than 150 million cases of UTI are reported annually, and the numbers have been increasing [2]. The UTIs are classified as upper UTIs (pyelonephritis) and lower UTIs (cystitis, prostatitis), depending on the site of infection, complication due to underlying diseases, and anatomical function. The UTIs may present febrile and non-specific signs and symptoms, which can cause a delay in treatment and serious complications leading to hypertension, chronic renal insufficiency, and end-stage renal failure. Moreover, they are significant morbidity factors for both outpatients and inpatients [1,3]. Among hospital-acquired (nosocomial) infections, UTIs are the most commonly caused by using an indwelling urinary catheter [4]. Nosocomial infections contribute to severe economic and public health issues for healthcare institutions [4]. UTIs can be missed due to inaccurate diagnosis, resulting in poor treatment and prognosis despite the availability of international recommendations for the management of UTIs, advances in diagnostic technology, care improvements, and innovative treatments [5]. Moreover, clinical presentation, etiology, and antimicrobial susceptibility patterns vary by region [6]. A clear understanding of UTI symptoms and treatment approaches is crucial for appropriate antibiotic choice, preventing severe complications and antibiotic misuse, and inhibiting the development of antibiotic-resistant bacteria [6]. The most common pathogen isolated from urine cultures is *Escherichia coli*, 80–90% [7]. However, other bacteria that were rarely isolated previously are now rising *(Proteus*, *Citrobacter*, *Enterobacter*, and *Serratia* species) [8]. *E. coli* can produce extended-spectrum β-lactamase (ESBL) enzymes, which provide resistance against drugs like penicillins, extended-spectrum cephalosporins, and monobactams. These ESBL-producing bacteria are associated with increased morbidity and mortality due to inappropriate antibiotic regimens [9]. The rising occurrence of UTIs and the inappropriate usage of antibiotics necessitate employing the correct regimen based on the local antimicrobial sensitivity of different bacteria [10]. Resistance to beta-lactam antibiotics and trimethoprim-sulfamethoxazole (TMP-SMX) has increased in microbes infecting pediatric patients [5]. Several variables must be taken into account while selecting antibiotics. The antimicrobial susceptibility of a given microbe might vary with habitat and with time, making it one of the most significant factors to consider. Hence, routine and widespread antibiotic susceptibility tests are necessary [10]. This variation in susceptibility makes it challenging for clinicians to develop standardized guidelines for the treatment of UTIs. Therefore, a systematic approach to the management of UTI is indispensable [11,12]. 

Despite increased antibiotic resistance, beta-lactam antibiotics and TMP-SMX remain first-choice treatments for pediatric UTIs [8]. Patients diagnosed with UTI are administered empirical antimicrobial treatment until the etiological agent is determined, which may further increase antibiotic resistance [7]. Our study aimed to determine the antimicrobial susceptibility patterns of pediatric UTI-causing pathogens, including ESBL-producing bacteria, in Saudi Arabia. This is the first study in the region to assess ESBL-producing bacteria specifically in the pediatric age group. Conducted during the COVID-19 period, our findings have significant implications for understanding the impact of the pandemic on healthcare outcomes. By providing valuable insights into antimicrobial resistance and appropriate antibiotic prescribing, our results can guide empirical treatment decisions, improve patient outcomes, and mitigate the consequences of antibiotic misuse.

## 2. Materials and Methods

### 2.1. Study Setting

This research was carried out at King Fahad Medical City, a large tertiary hospital in Riyadh, Saudi Arabia. The local health centers, private hospitals, and clinics refer patients to this hospital. The Institutional Review Board of KFMC approved this study (IRB 00010471) and waived the requirement of obtaining informed consent from the participants owing to the retrospective nature of the study.

### 2.2. Patients and Bacterial Isolates

The study involved a retrospective, cross-sectional analysis of all urine samples delivered to the microbiology lab for culture and sensitivity testing between 1 January 2019 and 31 December 2020, from both inpatients and outpatients suspected of having a UTI. Using a standard data collection sheet, demographic data, clinical data (patient history, comorbidities, duration of therapy, recurrence, and immunological state), the organism isolated, and antimicrobial sensitivity profiles were obtained. Patients with a history of fever > 38 °C or symptoms of UTIs including dysuria, hesitancy, frequency, urgency, low-volume voids, or lower abdomen discomfort met the inclusion criteria of the study. In infants and children, a urinary tract infection was defined as the growth of a single pathogen of >10 × 5 colony-forming units/mL (CFU/mL) in a correctly obtained urine specimen (suprapubic aspiration, transurethral catheterization, or mid-stream urine) from children with fever or other urinary symptoms [13].

The study implemented rigorous criteria for sample selection to ensure accuracy. Urine samples were excluded if they had multiple types of bacteria or were collected more than three days after hospital admission. A meticulous registration process and adherence to European guidelines ensured proper documentation and the transfer of samples to the microbiological laboratory. In cases where rapid transportation was not feasible, appropriate measures were taken to maintain sample integrity by refrigerating them at controlled temperatures between 4 and 6 °C [14,15]. This ensured that sample quality was preserved during transit. Specific factors, such as clinical practice guidelines for urinary tract infection (UTI) and excluding samples associated with chronic renal failure, severe birth defects, and pediatric bags, were considered to minimize contamination and confounding factors. By following these stringent procedures, the study aimed to provide reliable insights into resistance rates and patterns within the cohort, upholding scientific rigor and validity.

### 2.3. Microbiological Culturing of Urine Samples: Stringent Selection and Methodological Approaches

The urine samples underwent culturing using a semiquantitative technique based on the World Health Organization (WHO) guidelines, utilizing standard culture media [16]. To provide a brief overview, 1 mL of urine was streaked onto cystine-lactose-electrolyte-deficient (CLED) and blood agar plates (Hardy Diagnostics, Santa Maria, CA, USA). The plates were then incubated aerobically at 37 °C for 24 h using a calibrated wire loop. The isolates were subsequently characterized and identified using biochemical tests and cultural traits. The BD Phoenix (BD-Canada, Mississauga, ON, Canada) and API ID (bioMerieux UK Ltd., Basingstoke, UK) systems were employed for this purpose.

### 2.4. Utilization of the BD Phoenix System for Identification of Bacterial Isolates

The NMIC/ID-4 panel of the BD Phoenix system was employed to analyze a total of 1022 bacterial isolates. Initially, a McFarland 0.5 stock solution was prepared in 4.5 mL of Phoenix ID broth [17] using a Sensi Titre™ nephelometer (Thermo Fisher Scientific, Waltham, MA, USA). The resulting bacterial ID solution was then placed in the designated ID field on the Phoenix panel. Subsequently, 50 µL of the microbial solution was added to each chemical reactivity well on the panel. Any excess suspension was absorbed by the cushion located on the underside of the panels. The panel was sealed with a rubberized coating, and the display code was read before being directly inserted into the Phoenix machine.

For the cultivated bacterial isolates, the ID results obtained from the Phoenix system were compared with those of the traditional API system, which served as the reference standard. Accurate identification rates were subsequently determined at both the genus and species levels based on these comparisons.

### 2.5. Antimicrobial Susceptibility Testing of Escherichia coli Strains using the Kirby Bauer Method

According to the recommendations of the Clinical Laboratory Standard Institute (CLSI) version 6.0, all identified bacterial isolates were subjected to in vitro investigation of their antimicrobial sensitivity using the agar disc diffusion technique (Hardy Diagnostics, Santa Maria, CA, USA) [18]. The following antibiotics were included in our research: ampicillin (10 μg), amoxicillin-clavulanate (20/10 μg), cephalothin (30 μg), cefuroxime (30 μg), ceftazidime (30 μg), cefoxitin (30 μg), cefepime (30 μg), cefotaxime (30 μg), ceftriaxone (30 μg), ciprofloxacin (5 μg), gentamicin (10 μg), amikacin (30 μg), trimethoprim-sulfamethoxazole (1.25/23.75 μg), piperacillin-tazobactam (36 μg), imipenem (10 μg), meropenem (10 μg), ertapenem (10 μg), levofloxacin (5 μg), tigecycline (15 μg), nitrofurantoin (300 μg), and colistin (10 μg).

The experimental analysis of the obtained results was performed in accordance with the CLSI’s suggested diameters or breakpoints [18], determining whether the isolates were susceptible (S), intermediate (I), or resistant (R) to the tested antibiotics. In cases where an isolate demonstrated resistance to three different antibiotic groups, it was classified as multidrug-resistant. By employing the Kirby Bauer method, the study aimed to evaluate the susceptibility patterns of *E. coli* strains to a wide range of antibiotics, considering the CLSI guidelines, and identify potential multidrug-resistant isolates.

### 2.6. Application of the Double-Disc Synergy Test (DDST) for the Detection of Extended-Spectrum Beta-Lactamase (ESBL) Activity

ESBL detection in bacterial pathogens was conducted using ceftazidime (30 μg), cefotaxime (30 μg), cefpodoxime (30 μg), and amoxicillin/clavulanic acid (amoxicillin 20 μg + clavulanic acid 10 μg) [19]. The DDST (Double Disc Synergy Test) method was employed for this purpose. On Muller–Hinton agar (MHA) plates, the amoxicillin/clavulanic acid disc (20 μg/10 μg) and discs containing third-generation cephalosporins were placed at a distance of approximately 20 mm from center to center. Following overnight incubation at 37 °C, any increase in the inhibition zone of the cephalosporin discs towards the amoxicillin/clavulanic acid disc indicated a positive ESBL finding.

### 2.7. Statistical Analyses

The SPSS statistical program version 25 (IBM Inc., Armonk, NY, USA) was used to analyze the data. The data was divided according to age and sex, and the prevalence of the various uropathogens and the patterns of their resistance were demonstrated as percentages. The normality of the distribution was assessed using the Kolmogorov–Smirnov test. The chi-square test was used to compare the different categories. Two-tailed *p* < 0.05 was considered a statistically significant difference.

## 3. Results

### 3.1. Sociodemographic Characteristics 

This study was conducted during the period 2019–2020, and the study included 1022 patients after excluding 172 urine cultures due to exclusion criteria. A total of 383 (37.5%) of these patients were males, and 639 (62.5%) were females. Of these patients, 39 (3.8%), 253 (24.8%), and 730 (71.4%) were neonates (<28 days), infants (28 days to 1 year), and children (1 year to 15 years), respectively. The predominant causative agents were *E. coli* (58.6%), *Klebsiella* sp. (23.9%), *Pseudomonas* sp. (7.1%), *Proteus* sp. (5%), *Enterobacter* sp. (3.5%), and *Citrobacter* sp. (1.3%), and the least common bacteria was *Serratia* sp. (0.6%). Analysis of the results according to patient sex revealed that, although *E. coli* was the predominant isolated pathogen in both sexes, it occurred more frequently in females (69.8%) compared with males (39.9%) (significant at *p* = 0.000; chi-square = 68.73), whereas the prevalence of UTI due to *Klebsiella* sp. was higher in males (34.5%) compared with females (17.5%) (significant at *p* = 0.004; chi-square = 11.11). The prevalence of UTI caused by *Pseudomonas* sp. was significantly higher in males (12.3%) compared with females (4.1%) (significant at *p* = 0.024; chi-square = 7.46) (Table 1).

### 3.2. Antimicrobial Susceptibility Patterns

Table 2 presents the sensitivity of pathogens to antibiotics isolated from the urinary tract. *E. coli* isolates were more sensitive to meropenem and ertapenem in 99.2% of cases, followed by amikacin (99%), nitrofurantoin (96.8%), and imipenem (95.2%). While *Klebsiella* sp. was sensitive to amikacin in 97.1%, followed by meropenem and ertapenem in 92.2%. Among the antibiotics used for sensitivity testing of the isolates, meropenem, ertapenem, and piperacillin-tazobactam showed the highest activity (100%) against *Proteus* sp. and *Citrobacter* sp. The *Enterobacter* sp. isolates were more sensitive to amikacin (100%), and meropenem and ertapenem (94.4% in both), whereas the *Pseudomonas* sp. were more sensitive to amikacin at 93.2%, followed by cefepime and gentamicin (87.7% in both). In general, most pathogens were sensitive to amikacin (98%), followed by ertapenem (97.2%) and meropenem (96.2%).

### 3.3. Susceptibility Patterns among ESBL and Non-ESBL Uropathogens

ESBL isolates showed high sensitivity to meropenem and ertapenem (100% in both), followed by amikacin (99%), imipenem (97%), and cefoxitin (90%). Moreover, the ESBL isolates demonstrated a very weak sensitivity towards the antimicrobials cephalothin, cefuroxime, cefotaxime, ceftriaxone (0% in all), and (1%) in cefepime and ceftazidime. Non-ESBL isolates were markedly more sensitive to the majority of the antimicrobials and tested 98% in amikacin, followed by meropenem and ertapenem (96% in both), cefepime (94%), ceftazidime (93%), and gentamicin (92%). In general, ESBL and non-ESBL were found to be more sensitive to cefoxitin, amikacin, meropenem, and ertapenem antimicrobials. (Figure 1).

## 4. Discussion

This study presented the prevalence and antibiotic sensitivity pattern of uropathogenic bacteria in pediatric patients with UTI at a tertiary care center in KFMC during the period 2019–2020. Our study involved 1022 patients and demonstrated that *E. coli* was the most common etiologic factor for UTIs in children under the age of 15. This is consistent with previous studies, which reported a frequency of 58%, 65.2%, 64.2%, and 75.7% [7,10,20]. However, in the study by Heidary et al., *P. aeruginosa* was reported as the most common pathogen (49.65%) [1]. In addition, our study revealed that the prevalence of *E. coli* among females was significantly higher than in males, whereas the prevalence of other uropathogens such as *Klebsiella*, the second most common pathogen, and *Pseudomonas* sp. was significantly higher in males than in females. These findings are consistent with previous studies [20,21,22]. Regarding the prevalence of pathogens among age categories, the chi-square test result of our study showed significant differences (*p* = 0.000; chi-square = 60.496). *E. coli* was higher in children (64.1%), the incidence of *Klebsiella* was higher in infants (37.5%), and that of *Citrobacter* was higher in neonates (2.6%). However, in the study by Seyed Reza et al., the *E. coli* prevalence was reported to be higher in infants (63.1%), that of *Klebsiella* was higher in neonates (32.9%), and that of *Pseudomonas aeruginosa* was higher in infants [20]. Therefore, as demonstrated in the study by Afsharpaiman et al., age may influence the etiology of UTIs [23]. Hence, these findings highlight the importance of considering the patient’s age and sex when administering empirical treatment. 

Regarding antimicrobial susceptibility, our study revealed that *E. coli* isolates were more sensitive to meropenem and ertapenem in 99.2% of cases, followed by amikacin (99%), nitrofurantoin (96.8%), and imipenem (95.2%). These findings are in agreement with previous studies [24,25,26,27]. And in disagreement with the study by Shar-iian et al. in 2006 in Tehran, which reported that ceftriaxone (97.8%) and cefotaxime (95.2%) were the most sensitive antibiotics for *E. coli* [26].

In contrast, the least sensitive antibiotics for *E. coli* in our study were found to be ampicillin (19.7%), cephalothin (40.3%), and trimethoprim-sulfamethoxazole (51.8%). These results appear to be inconsistent with the results of previous studies [22,28,29].

In the case of *Klebsiella* sp., the highest antimicrobial sensitivity rates in our study were found for amikacin in 97.1% of cases, followed by meropenem and ertapenem (92.2% of cases). Similarly, in a previous local retrospective study conducted in Abha, Saudi Arabia, the highest antimicrobial sensitivity rate in *Klebsiella* sp. was found for meropenem (100%), tigecycline (81.82%), and amikacin (72.73%) [30].

Overall, in our study, the isolates showed high sensitivity to amikacin (98%), followed by ertapenem (97.2%) and meropenem (96.2%). Similarly, in a previous local retrospective study conducted in Unaizah, Saudi Arabia, high sensitivity for isolates was observed for carbapenem (meropenem/imipenem) (100%) in both [22]. Whereas, in a study from Khartoum, Ali and Osman [31] reported that all isolates were highly sensitive to gentamicin (96%), ciprofloxacin (94%), and ceftriaxone (90%).

To the best of our knowledge, there are no reported data in Saudi Arabia regarding ESBL antimicrobial susceptibility in the pediatric age group. Our study demonstrates that ESBL-producing bacteria have lesser antimicrobial susceptibility to most antimicrobials compared to non-ESBL. The highest sensitivity of antimicrobials toward ESBL was demonstrated by meropenem and ertapenem (100% in both), followed by amikacin (99%), which was a bit higher compared to the reports of other studies from Jordan, Turkey, and Nepal [10,32]. The higher sensitivity of amikacin in our study could be due to the very low prescription compared to other regions. In general, the antibiotic of choice against ESBL-producing bacteria is carbapenem, which agrees with our results [33].

Nitrofurantoin showed an 82% sensitivity against ESBL, which is similar to that of a study from Turkey [10], which was 80%. However, it was significantly higher than that reported in a study from Jordan (47.3%) [32]. Nitrofurantoin is a second-line drug in the treatment of ESBL infections; it is administered orally and demonstrates good absorption. However, its required dosing of q6H is considered a disadvantage. Despite the extensive usage of nitrofurantoin in our region, it still demonstrates good sensitivity against ESBL-producing bacteria [33].

Oral treatment options for UTIs include cefepime (68.1%), amoxicillin with clavulanic acid (54%), trimethoprim-sulfamethoxazole (51.8%), nitrofurantoin (79%), and ciprofloxacin (75%). For empirical treatment, locality-wise tests of antibiotic susceptibility to commonly prescribed antibiotics should be conducted due to the significant difference in susceptibility between nations. Therefore, *Coliforms* sp. should be tested against the most commonly prescribed antibiotics, namely, TMP-SMX and cephalexin.

Regarding the treatment of febrile infants with UTI (pyelonephritis) or its consequence (urosepsis). The preferred antibiotic should have a high or therapeutic concentration in the bloodstream to be effective. Nitrofurantoin is an example of an antibiotic that does not reach its therapeutic concentration in the bloodstream and is primarily eliminated in the urine. Consequently, it should not be used to treat febrile infants with urinary tract infections (UTIs) [10].

### 4.1. Antimicrobial Susceptibility Patterns in COVID-19 Era

Our study findings were consistent with previous research regarding antimicrobial susceptibility patterns, indicating agreement with existing literature. Furthermore, our analysis aligns with studies that have examined the impact of COVID-19 on antimicrobial sensitivity [34,35]. Although we did not observe any significant differences in susceptibility patterns that could be attributed to the pandemic, it is important to note that further quantitative studies are warranted for comparative analysis. Future investigations can provide a more comprehensive understanding of any potential quantitative differences that may arise.

### 4.2. Strengths and Limitations

This study contributes significantly to the field by addressing the lack of local data on antimicrobial susceptibility patterns in the pediatric age group, as recommended by the World Health Organization (WHO) in its Global Action Plan to combat antimicrobial resistance (AMR). Notably, it is the first study conducted in Saudi Arabia to assess the antimicrobial susceptibility patterns of ESBL-producing bacteria specifically in the pediatric population. Additionally, it is the first study to investigate antimicrobial susceptibility patterns during the COVID-19 pandemic.

Several limitations need to be acknowledged. Firstly, the retrospective nature of the study introduces inherent limitations associated with this study design. Secondly, while our study population represents patients from diverse regions in Saudi Arabia due to KFMC’s admissions, the results may have limited generalizability as they originate from a single tertiary center. Thirdly, certain cases might have been overlooked, such as those receiving treatment without a urine culture or prior antibiotic administration, and a limited number of non-*E. coli* bacteria were included due to their lower frequency rates. Furthermore, given that the study was conducted at the beginning of the COVID-19 pandemic and spanned only two years, it becomes challenging to identify significant differences in susceptibility patterns during this specific period. Moreover, we did not compare sensitivity patterns before and during the pandemic to assess any potential differences. Lastly, it is important to acknowledge the potential for bias as the samples were collected solely from a single tertiary center.

Despite these limitations, our study significantly contributes to the understanding of antimicrobial susceptibility patterns in the pediatric population, aligning with WHO recommendations to combat AMR. The findings serve as a valuable reference for guiding clinical decision-making and formulating local strategies to combat antimicrobial resistance in Saudi Arabia and beyond. Future research endeavors should focus on larger-scale studies with diverse patient populations and longitudinal analyses to capture a more comprehensive assessment of susceptibility patterns and potential variations before and during the pandemic.

## 5. Conclusions

Based on our study findings, we recommend that empirical antibiotic selection for UTIs be informed by local antibiotic sensitivity data and the prevalence of bacterial species rather than relying solely on global or national standards. Our study identified amikacin, ertapenem, and meropenem as the most effective intravenous antibiotics for empirical therapy against UTI-causing pathogens, including ESBL-producing bacteria. For oral treatment, suitable options include cephalosporin, cefuroxime, or amoxicillin plus clavulanic acid. Oral nitrofurantoin may also be a viable choice to address both ESBL and non-ESBL UTIs in pediatric patients. It is important to note that the decision regarding empirical therapy should be made by specialized physicians, taking into consideration the patient’s age and sex. By implementing these recommendations, our study aims to enhance patient management and treatment outcomes.

Furthermore, our study did not observe any changes in antimicrobial susceptibility patterns that could be attributed to the COVID-19 pandemic when compared to previous studies. However, it is crucial to conduct future studies comparing susceptibility patterns before and during the pandemic to assess for any noteworthy alterations. This comparative analysis would provide valuable insights into the potential impact of the pandemic on antimicrobial resistance.

## Figures and Tables

**Figure 1 children-10-00971-f001:**
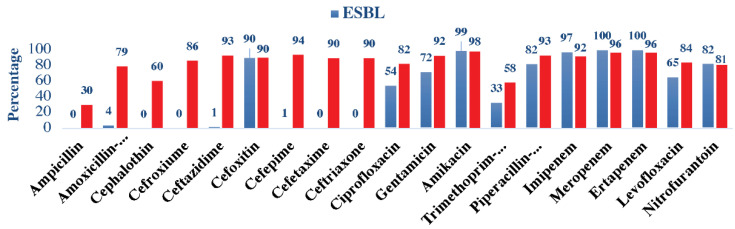
Antimicrobial susceptibility patterns among ESBL and non-ESBL uropathogens.

**Table 1 children-10-00971-t001:** Distribution of uropathogens causing urinary tract infections according to age and gender.

AGE	Neonates (3.8%)			Infants (24.8%)			Children (71.4%)			Pearson	
**Bacteria**	**Total** **(N = 39)**	**Male** **(*n* = 22)**	**Female** **(*n* = 17)**	**Total** **(N = 253)**	**Male** **(*n* = 155)**	**Female** **(*n* = 98)**	**Total** **(N = 730)**	**Male** **(*n* = 206)**	**Female** **(*n* = 524)**	**Chi-Square**	** *p* **
*E. coli*	59.0	59.1	58.8	42.7	36.8	52.0	64.1	40.3	73.5	68.729	0.000
*Klebsiella* sp.	25.6	22.7	29.4	37.5	41.3	31.6	19.0	30.6	14.5	11.114	0.004
*Proteus* sp.	0.0	0.0	0.0	3.6	3.2	4.1	5.8	8.3	4.8	1.687	0.407
*Enterobacter* sp.	2.6	4.5	0.0	3.2	1.3	6.1	3.7	7.3	2.3	3.333	0.189
*Citrobacter* sp.	2.6	4.5	0.0	1.2	1.3	1.0	1.2	2.4	0.8	3.001	0.221
*Serratia* sp.	2.6	0.0	5.9	1.6	1.9	1.0	0.1	0.0	0.2	3.000	0.223
*Pseudomonas* sp.	7.7	9.1	5.9	10.3	14.2	4.1	6.0	11.2	4.0	7.462	0.024

**Table 2 children-10-00971-t002:** Antibiotic sensitivity pattern of common uropathogens in urinary tract infections.

Antibiotic	*E. coli*	*Klebsiella*	*Proteus* sp.	*Enterobacter* sp.	*Citrobacter* sp.	*Serratia*	*Pseudomonas* sp.	Overall Sensitive
	(*n* = 599)	(*n* = 244)	(*n* = 51)	(*n* = 36)	(*n* = 13)	(*n* = 6)	(*n* = 73)	
Ampicillin	28.4	0	33.3	0	0	0	NT	19.7
Amoxicillin-Clavulanate	54.4	55.3	86.3	0	53.8	0	NT	54.0
Cephalothin	36.7	49.6	74.5	0	23.1	0	NT	40.3
Cefuroxime	61.6	54.1	80.4	5.6	38.5	0	NT	57.9
Ceftazidime	65.4	59.4	94.1	63.9	92.3	50.0	86.3	67.1
Cefoxitin	91.7	84.4	100.0	0	38.5	0	NT	85.5
Cefepime	66.1	60.2	96.1	72.2	84.6	50.0	87.7	68.1
Cefotaxime	64.4	56.6	80.4	47.2	76.9	16.7	NT	62.5
Ceftriaxone	64.4	56.6	80.4	47.2	76.9	16.7	NT	62.5
Ciprofloxacin	71.3	80.3	76.5	83.3	92.3	100.0	84.9	75.5
Gentamicin	89.0	81.1	74.5	88.9	100.0	33.3	87.7	86.1
Amikacin	99.0	97.1	96.1	100.0	100.0	100.0	93.2	98.0
Trimethoprim-Sulfamethoxazole	47.6	59.4	43.1	63.9	84.6	100.0	NT	51.8
Piperacillin-Tazobactam	94.3	76.6	100.0	72.2	100.0	50.0	84.9	88.7
Imipenem	95.2	88.9	92.2	91.7	100.0	83.3	80.8	92.4
Meropenem	99.2	92.2	100.0	94.4	100.0	83.3	83.6	96.2
Ertapenem	99.2	92.2	100.0	94.4	100.0	83.3	NT	97.2
Levofloxacin	73.8	90.2	76.5	91.7	92.3	100.0	74.0	78.9
Nitrofurantoin	96.8	59.4	2.0	44.4	61.5	NT	NT	79.8

## Data Availability

Data will be made available upon reasonable request directed to ibraheemaltamimi02@gmail.com.

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
