# Peer review of "Bacterial Pathogens and Antimicrobial Susceptibility Patterns of Urinary Tract Infections in Children during COVID-19 2019–2020: A Large Tertiary Care Center in Saudi Arabia"

_children, 2023, doi:10.3390/children10060971_

Round 1
Reviewer 1 Report
Major Comments:
1. Please describe the importance of the study in the Introduction?
2. Results: 3.1: Why UTI cases are more in female child than male child? Can the authors explain this?
3. How did the authors test sensitivity and resistance of pathogens to antibiotics? What were the criteria or parameters considered here? Please clarify in the ‘Methods’ section.
4. How genders influence on the sensitivity and resistance of each bacterial pathogen to the different antibiotics? Did the authors observe any correlation?
5. Why didn’t the authors consider adult UTI patients in their study? Can the authors predict about the sensitivity and resistance patterns of the bacterial pathogens to the tested antibiotics in adult UTI patients? Please discuss.
6. How Covid-19 is related or influence to the sensitivity and resistance patterns of the bacterial pathogens to the tested antibiotics? The authors didn’t discuss much about the impact/relation of Covid-19 on/with their study outcomes?
Minor Comments:
1. Results: 3.1: line:1: 2021–2020??
2. The manuscript should be checked for the grammatical errors and typos.
The manuscript should be checked for the grammatical errors and typos.
Author Response
We sincerely appreciate the valuable comments and thorough peer review provided by the reviewer. We would like to express our gratitude for the constructive feedback, which has greatly contributed to the improvement of our paper. We have carefully considered and addressed each point raised, making significant changes accordingly. Your input has been instrumental in enhancing the quality and rigor of our study.
Note: Attached the manuscript after extensive revision and per the reviewers comments
Please describe the importance of the study in the Introduction?
Response: We have revised and rewrote the rationale of the study to emphasize its significance and contribution to the existing literature. The revised introduction highlights the importance of addressing the knowledge gap in antimicrobial stewardship.
Why UTI cases are more in female child than male child? Can the authors explain this?
Response: Urinary tract infection (UTI) exhibits a higher prevalence among females, primarily attributed to anatomical differences. Females possess a relatively shorter urethra, which increases their susceptibility to infections, even in adulthood. This anatomical characteristic creates a favorable environment for microbial colonization and subsequent infection.
How did the authors test sensitivity and resistance of pathogens to antibiotics? What were the criteria or parameters considered here? Please clarify in the ‘Methods’ section
Response: Response: We have made significant revisions to the methodology section to address these inquiries. The revised version provides detailed information on the methods employed for testing the sensitivity and resistance of pathogens to antibiotics. It encompasses the specific criteria and parameters considered during the evaluation process. The updated 'Methods' section offers a comprehensive overview of the experimental approach and provides clarity regarding the testing procedures and assessment criteria employed in the study.
How genders influence on the sensitivity and resistance of each bacterial pathogen to the different antibiotics? Did the authors observe any correlation?
response:
Reviewer: How does gender influence the sensitivity and resistance of each bacterial pathogen to different antibiotics? Did the authors observe any correlation?
Response: We acknowledge the comment and recognize that gender may potentially influence the sensitivity patterns. However, due to the extensive data and numerous antibiotics analyzed, it was challenging to directly compare sensitivity patterns between genders. This limitation is duly noted in our study.
Why didn’t the authors consider adult UTI patients in their study? Can the authors predict about the sensitivity and resistance patterns of the bacterial pathogens to the tested antibiotics in adult UTI patients? Please discuss.
Response: Response: The decision to focus our study on pediatric patients was driven by the limited research available on this age group in our region and globally. Adult antimicrobial sensitivity patterns, including UTIs, have been extensively studied, particularly in our region. Therefore, our study specifically aimed to address the knowledge gap in pediatric UTIs
How Covid-19 is related or influence to the sensitivity and resistance patterns of the bacterial pathogens to the tested antibiotics? The authors didn’t discuss much about the impact/relation of Covid-19 on/with their study outcomes?
We appreciate the reviewer's comment regarding the impact of COVID-19 on the sensitivity and resistance patterns of bacterial pathogens to the tested antibiotics. In response to this concern, we have revised our study to include a more comprehensive discussion on the potential influence of COVID-19 on antimicrobial changes. We have incorporated relevant information in the introduction, discussion, conclusion, and abstract sections to address this important aspect. Furthermore, it is important to note that our study did not include a comparison between the pre-COVID-19 and during-COVID-19 periods, which would have provided quantitative insights into any significant differences. This limitation has been duly acknowledged and incorporated into the limitations section of our study.
- Results: 3.1: line:1: 2021–2020??
- The manuscript should be checked for the grammatical errors and typos.
Response: We express our gratitude for your valuable time and effort in reviewing our paper. We have thoroughly reviewed and addressed the errors, refined the English language usage throughout the manuscript, and incorporated additional references to enhance the methodology section, as per your insightful comments. Your feedback has been instrumental in improving the quality of our work, and we sincerely appreciate your contribution.

Author Response
We sincerely appreciate the valuable comments and thorough peer review provided by the reviewer. We would like to express our gratitude for the constructive feedback, which has greatly contributed to the improvement of our paper. We have carefully considered and addressed each point raised, making significant changes accordingly. Your input has been instrumental in enhancing the quality and rigor of our study.
Note: Attached the manuscript after extensive revision and per the reviewers comments
Reviewer: Comments on abstract
Response : The abstract has been thoroughly revised based on the provided comments, encompassing the objectives, methodology, and conclusion of the study.
Reviewer: The rationale of the study should be modified.
Response: The rationale of the study has been rewritten.
Reviewer: To gain a better understanding of the work, more information is necessary. This can be achieved through :
1. Providing additional details in this section for greater clarity.
2. Citing references that support the methodology used.
Response: Our research team has extensively revised the methodology section of the study, incorporating additional details to provide a comprehensive account of how the research was conducted. Furthermore, six references have been included to augment the rigor and quality of the work. This ensures a more formal and cohesive presentation of the research methodology.
Reviewer: Improving the analysis of data will be beneficial to the paper and results section.
Response: While acknowledging that further analysis could have been beneficial in the results section, we recognize the challenge posed by the extensive data and large sample size, which made it difficult to incorporate additional analysis. Implementing such analysis would undoubtedly have enhanced the outcomes and overall quality of the paper
Reviewer: how do you get this? . Susceptibility Patterns among ESBL and Non-ESBL Uropathogens
Response: In our study, we conducted a comparative analysis of the sensitivities of antimicrobials against uropathogens that secrete Extended-Spectrum Beta-Lactamase (ESBL) and those that do not secrete ESBL. The objective was to identify any antimicrobial agent that exhibits efficacy in eradicating uropathogens that secrete ESBL, irrespective of their bacterial species. While the diverse range of bacteria could potentially pose a limiting factor, our objective remained focused on identifying an antimicrobial agent with efficacy against ESBL-secreting uropathogens, regardless of the specific bacterial pathogen.
Reviewer: Reviewer: How did you conduct a compare antimicrobial sensitivities across different studies and determine whether there was agreement or disagreement among them?
Response: Our approach primarily relied on identifying antimicrobials that exhibited the highest and closest sensitivity values to our study findings. For instance, in Saudi Arabia, we observed that meropenem and etrapenem were consistently reported as the most sensitive antimicrobials in other studies as well. In cases where variations in numerical values arose, we thoroughly discussed these differences in the subsequent discussion section. Nonetheless, our main criterion for ordering antimicrobials was based on their ranking in terms of highest sensitivity.
Reviewer: Giving the reference number after the name (s) will be better.
Response: Thank you for your valuable feedback the change has been made
Reviewer: It is impossible to draw a definitive conclusion based on an in vitro study. It is best to avoid being too definitive in this case.
Response: We have changed accordingly.
We express our sincere gratitude for the time and effort you have dedicated to providing valuable feedback on our paper. Your insights have proven to be immensely valuable. We extend our best wishes to you as well.

Reviewer 3 Report
The topic under study is undoubtedly an important issue in the daily routine of both hospital and area pediatricians, since urinary tract infections are an increasingly common pathology and the management of antibiotic resistance is becoming increasingly important. However, it must be emphasized that this work does not involve any significant changes to current clinical practice, both in terms of epidemiology (pathogen identification is in line with European and American classics) and therapy. The molecules cited as therapeutic solutions are, in order of efficacy, those already in common use.
The limitations of the study, such as retrospectivity, cited by the authors themselves, make the quality of the study poor. However, the methods are accurately described, and the results are clearly stated and consistent with the hypotheses adopted.
It should be acknowledged that the study conducted is a study with a large sample size, although no stratification between risk classes (neonates, pediatric patients, hospitalized or not) was performed, and that it is the first study on the prevalence of antibiotic susceptibility patterns of uropathogenic bacteria in pediatric patients with UTI in Saudi Arabia
Author Response
We sincerely appreciate the valuable comments and thorough peer review provided by the reviewer. We would like to express our gratitude for the constructive feedback, which has greatly contributed to the improvement of our paper. We have carefully considered and addressed each point raised, making significant changes accordingly. Your input has been instrumental in enhancing the quality and rigor of our study.
We would like to highlight the significant revisions made in our paper, including the following key changes:
- Rationale: We thoroughly revised and improved the rationale of the study to enhance its clarity and relevance.
- Methodology: The methodology section was extensively edited, addressing concerns raised about the need for additional details and references. We ensured that the cited references were appropriately improved.
- Consideration of COVID-19: We incorporated comprehensive information on the impact of COVID-19 on antimicrobial sensitivity, taking into account the relevant aspects and implications.
- Limitations and strengths: We carefully reworded and refined the limitations and strengths section, providing a more accurate and comprehensive assessment.
- Conclusion: The conclusion section was completely rewritten, emphasizing the key findings and their implications.
- Abstract: We revised the abstract to accurately summarize the main aspects of the study.
These major changes significantly enhance the overall quality and value of our paper.
Once again we thank you for your time and effort attached is the updated manuscript

Round 2
Reviewer 1 Report
The authors need to highlight the changes made in the revised manuscript for evaluation.
Reviewer 2 Report
The manuscript can be published.